# Acute Erythroid Leukemia: From Molecular Biology to Clinical Outcomes

**DOI:** 10.3390/ijms25116256

**Published:** 2024-06-06

**Authors:** Priyanka Fernandes, Natalie Waldron, Theodora Chatzilygeroudi, Nour Sabiha Naji, Theodoros Karantanos

**Affiliations:** 1Johns Hopkins School of Medicine, Baltimore, MD 21205, USA; pferna15@jhmi.edu (P.F.); nwaldro2@jhmi.edu (N.W.); 2Department of Oncology, Johns Hopkins School of Medicine, Baltimore, MD 21205, USA; tchatzi1@jh.edu (T.C.); nnaji1@jh.edu (N.S.N.)

**Keywords:** Acute Erythroid Leukemia, Acute Myeloid Leukemia, myeloid neoplasms, hematopoietic disorders

## Abstract

Acute Erythroid Leukemia (AEL) is a rare and aggressive subtype of Acute Myeloid Leukemia (AML). In 2022, the World Health Organization (WHO) defined AEL as a biopsy with ≥30% proerythroblasts and erythroid precursors that account for ≥80% of cellularity. The International Consensus Classification refers to this neoplasm as “AML with mutated *TP53*”. Classification entails ≥20% blasts in blood or bone marrow biopsy and a somatic *TP53* mutation (VAF > 10%). This type of leukemia is typically associated with biallelic *TP53* mutations and a complex karyotype, specifically 5q and 7q deletions. Transgenic mouse models have implicated several molecules in the pathogenesis of AEL, including transcriptional master regulator GATA1 (involved in erythroid differentiation), master oncogenes, and CDX4. Recent studies have also characterized AEL by epigenetic regulator mutations and transcriptome subgroups. AEL patients have overall poor clinical outcomes, mostly related to their poor response to the standard therapies, which include hypomethylating agents and intensive chemotherapy. Allogeneic bone marrow transplantation (AlloBMT) is the only potentially curative approach but requires deep remission, which is very challenging for these patients. Age, AlloBMT, and a history of antecedent myeloid neoplasms further affect the outcomes of these patients. In this review, we will summarize the diagnostic criteria of AEL, review the current insights into the biology of AEL, and describe the treatment options and outcomes of patients with this disease.

## 1. Introduction

Acute Erythroid Leukemia (AEL) is a rare but aggressive subtype of Acute Myeloid Leukemia (AML) and constitutes 2% of all AML cases [1,2]. AEL is often associated with a complex karyotype, as well as the biallelic loss of *TP53*. The aggressive biology of this disease and its high-risk molecular features render AEL a particularly difficult disease to manage and cure [1]. As a result, the median survival of AEL ranges from 3 to 9 months, though studies incorporating the newest myeloid neoplasms classifications have noted that most AEL patients fall on the shorter end of that spectrum [3].

Most AEL cases arise in a de novo fashion, though a subgroup arises from antecedent myelodysplastic syndrome (MDS) or from other chronic myeloid neoplasms [2]. There are no definitive associations of genetic and environment risk factors with the development of AEL. A number of studies conducted before the newest myeloid diseases classifications have highlighted a possible implication of germline variants and environmental factors such as benzene toxin that could increase the risk of AEL [1,4].

In this review, we will discuss the various classification systems used in the definition of AEL, most notably the World Health Organization (WHO) and the International Census Classification (ICC) systems, and will describe the differing degrees of emphasis on disease etiology, aspirate percentages, and genetic markers [5,6,7]. Though classification systems differ in the naming convention of this disease, our review will use the most recent 2022 WHO naming terminology of “AEL” when discussing this disorder. Diagnostic features and work up will also be discussed, emphasizing the non-specific nature of clinical presentation, bone marrow aspirate features, flow cytometry markers that demonstrate erythroblast lineage, and additional genetic mutations linked to AEL. We will describe the current knowledge of the molecular biology of this disease, which is critical for the introduction of novel therapeutic strategies that are urgently needed for AEL. Finally, we will analyze the current therapeutic approaches for this disease, including chemotherapy, hypomethylating agents, and bone marrow transplantation.

## 2. Definitions

The condition referred to as erythroleukemia was first discovered in 1917 by Giovanni Di Guglielmo, who noted the large number of erythrocyte, platelet, and granule precursors. Di Guglielmo noted that this disorder could be subclassified as a pure acute, chronic, or mixed phenotype. In 1976, the French–American–British cooperative group defined this disease as AML-M6 based on the differentiation features of the leukemia cells [8]. In 2001, the WHO further categorized the FAB AML-M6 into the M6a and M6b subtypes. The M6a classification became known as the erythroid/myeloid leukemia subtype and required erythroblasts to comprise ≥50% of the total nucleated bone marrow cells and myeloblasts to compromise ≥20% of the remaining non-erythroid cells [8]. This subtype was removed in 2016 from the WHO classification system, and its previous criteria became encompassed by other subtypes, such as Myelodysplastic Syndrome and Myelodysplasia-Related Changes (MRC) AML. Contrastingly, the M6b subtype became known as Pure Erythroid Leukemia (PEL). This subtype required >80% erythroid precursors and ≥30% proerythroblasts in the bone marrow. There continues to be many classification criteria to redescribe this aggressive subtype of AML, most notably from the WHO and the ICC organizations.

### 2.1. World Health Organization

The WHO has revised their classification of AEL several times over the last few decades, making it more stringent with a higher cellularity and blast qualifying criteria. Their most recent 2022 classification defines AEL as a subset of differentiated AML that has (1) a bone marrow sample of ≥30% proerythroblasts and (2) pro-erythroid precursors that account for ≥80% of cellularity [5]. This definition can be challenging in practice, however, since many bone marrow biopsies in AEL are suboptimal and require cytology or immunohistochemistry to aid in diagnosis. As such, some cases with <80% cellularity are now being recognized within the AEL classification category [7]. This change is the main differentiating factor from the previous 2016 WHO criteria.

The WHO discusses the cellular features of AEL, noting maturation arrest, complex karyotype, and biallelic *TP53* mutations as supporters of the diagnosis. The presence of these findings does not change the classification label assigned to AEL. However, the findings of previous myeloid neoplasm/MDS or defining germline predispositions require a different and more specific classification of AEL [5]. Of note, the denominator used for calculating the blast percentage in myeloid neoplasms has changed to all nucleated BM cells, not just the “non-erythroid cells”, as in 2016 [9]. As a result, most cases previously diagnosed as erythroid leukemias are now classified as MDS with excess blasts.

### 2.2. International Census Classification

The 2022 International Census Classification (ICC) system emphasized both blast percentage and genomic factors when discussing the diagnosis of AEL/PEL and classifies this disorder within the broad category of myeloid neoplasms with mutated *TP53* due to their similarly aggressive clinical behavior [6]. One subcategory of this broad grouping is AML with mutated *TP53*. The criteria for this ICC diagnosis entail (1) ≥20% bone marrow or blood blasts or meeting the criteria for pure erythroid leukemia and (2) any form of a somatic *TP53* mutation with a variant allele frequency (VAF) >10% [6]. As determined by blast percentages, AEL/PEL, which commonly harbors a biallelic *TP53* mutation and a complex karyotype, now falls under the umbrella category of AML with mutated *TP53* [10]. It should be noted that the definition of AML with mutated *TP53* is broad and regularly includes non-AEL cases. It is also important to note that this classification is independent of the disease origin (de novo or secondary or treatment-related), unlike the 2022 WHO criteria.

## 3. Diagnosis

### 3.1. Presentation

The median age of AEL diagnosis is 67 years old, though some studies have demonstrated a bimodal age of diagnosis with a small peak at around 20 years old and a larger second peak in the early 70s [1,10]. This disease also demonstrates a slight male-to-female predominance (2.4:1) [11]. The clinical presentation of AEL can be non-specific in nature, as the most prominent symptoms and findings at diagnosis are fever and pallor, anemia (median hemoglobin of 7.5 g/L), hepatosplenomegaly, and evidence of hemolysis [1]. Clinicians should also ascertain a history of antecedent MDS, myeloproliferative neoplasm, and erythropoietin level, as this information impacts the diagnostic classification.

Diagnosis typically requires a bone marrow biopsy, though studies have raised the concern of the suboptimal quality of biopsies in AEL, as well as the frequent lack of blasts in peripheral blood, which makes the diagnosis challenging. Nevertheless, the most typical features in the bone marrow biopsy are hypercellularity, dyserythropoiesis, and a high percentage of erythroid precursors. Features frequently observed in the peripheral blood smear of AEL patients are leukopenia, basophilic stippling, and abnormal red blood cell morphology; however, these are not characteristic or diagnostic features of AEL [1].

### 3.2. Immunohistochemistry and Flow Cytometry

Immunohistochemistry and flow cytometry markers are frequently used for the better characterization of AEL cases. CD71 is a surface transferrin receptor that is present on most erythroid progenitors and is typically overexpressed in AEL blasts and erythroid malignant precursors [12]. Immature blasts in AEL may express Gerbich blood group (Gero) antigens, E-cadherin, carbonic anhydrase 1, CD36, and CD68 antigens, as well [1,7]. A dim expression of hemoglobin and Glycoprotein-A can also be found in these cells [1,13]. Additionally, other markers, such as GLUT1, have been shown to stain positively in cells of erythroid lineage [7,13]. It is important to note that Myeloperoxidase, HLA-DR, and CD33, which are known markers of myeloid lineage, are typically negative in the majority of cells in the biopsy [1,7,12]. CD13 and CD117 are highly variable among AEL cases, further highlighting some heterogeneity of this disease [7].

The differentiation pattern of cells based on phenotypic markers has been shown to be of prognostic value in AEL. Particularly, a higher percentage of proerythroblasts is associated with poorer AEL outcomes [3]. Additionally, a recent study showed that de novo AEL is associated with a better median survival (3.9 months) compared to patients with a history of MDS (2.6 months) or therapy-related AEL (2.3 months) [14]. The study notes that secondary AEL is characterized by a high incidence of refractory disease and early resistance to the current standard treatments, thereby leading to worse outcomes [15]. It should be highlighted that these conclusions are almost exclusively based on the former WHO/ICC categories.

### 3.3. Cytogenetic Characteristics

A complex karyotype is defined as at least three cytogenetic abnormalities and is an almost uniform feature of AEL [14]. In the setting of AEL, deletions in 5q and 7q, monosomy 5 and 7, and trisomy 8 are the most common abnormalities detected [1]. Additionally, a karyotypic abnormality that may be present on chromosome 17 (17p13) has been linked to the *p53* loss of function described in a significant percentage of AEL patients [10,16].

Cytogenetic variation amongst patients with AEL when stratified by age has been described. One study that focused on AEL cytogenetic characteristics examined 31 patients and found that patients under 45 years old had more cytogenic abnormalities compared to patients who were over 45 years old (66.7% vs. 54.5%) [17]. The significance of these results was mitigated by the small sample size.

Additionally, several studies have commented on the prognostic factors associated with each cytogenetic feature. t(8;21), t(15;17), inv(16), and del(20q) have all been associated with better outcomes among AEL patients [18]. Poorer outcomes are associated with the −5, −7, and abnormal 3q mutations. It is important to note, however, that most AEL patients have poor outcomes and that it is very challenging to identify chromosomal abnormalities that can potentially be linked with a better prognosis due to the severity and rarity of this disease. Many studies also contradict each other when discussing the prognosis associated with each of these alterations [1,19]. Moreover, the majority of these studies were conducted before the newest WHO/ICC changes, and as such, it may be difficult to extrapolate these results for the newest definitions. Regardless of this, most studies conclude that the absence of complex karyotype, even if it is very rare in AEL, could be associated with better outcomes and more prolonged survival [20,21].

## 4. Molecular Biology and Genomic Features of AEL

Based on the data from human samples and murine models, attempts have been made to elucidate the pathogenesis of AEL. Several potential mechanisms implicated in the pathobiology of AEL have been explored (Figure 1).

### 4.1. Erythropoietin Receptor (EPOR) Activation and Downstream JAK2 Signaling Pathway

Viruses inducing erythroleukemia phenotypes served as the first models to study the multistage nature of the disease [22]. The relation of constitutive EPOR activation with erythroleukemia pathogenesis was introduced early during the 1990s, as the erythroblastosis-inducing spleen focus-forming virus (SFFV) was found to activate EPOR [23], while activating point mutations of the *EPOR* also promoted tumorigenesis [24]. According to very recent data comparing the transcriptional data of human primary AEL tumors with other types of AML, EPOR is one of the upregulated genes in this type of leukemia [25]. Moreover, activation of the EPOR downstream signaling effectors, such as signal transcription activators (STATs), PI3K/AKT, and mitogen-activated protein (MAP) or extracellular signal-regulated (ERK) kinases [22,26,27], promote preleukemic development of proerythroblasts [26].

Genomic alterations in oncogenic signaling, such as the JAK2 and RAS pathways, have been described in AEL patients. Grossman et al. reported that only 3 out of 92 AEL patients carry mutations in *NRAS*, *KRAS*, or *FLT3* [20]. However, in a recent analysis by Takeda et al., gains and amplifications involving *EPOR*, *JAK2*, and/or *ERG/ETS2* were recurrently detected in AEL patients [28]. Additionally, another recent study found at least one of these mutations to be present in 5 of the 35 AEL cases [28]. The gains and amplifications of *EPOR* and *JAK2* were found to be more highly enriched in AEL than erythroid/myeloid leukemia cases [28]. These specific AEL cases showed enhanced cell proliferation, as well as sensitivity to ruxolitinib, in in vitro and xenograft models. As such, *JAK2* inhibition could be a potential therapeutic target in AEL in patients with *JAK2* gains and amplifications [28,29]. Similarly, several of these gene mutations were also found among the 41 AEL patients analyzed by the Mayo clinic group [10]. Particularly, 30% of the patients were found to have *JAK2* mutations, and 10% of the patients were found to have *NRAS* or *CSF3R* mutations, suggesting that kinase signaling pathways are possibly associated with the progression of AEL.

### 4.2. Erythroid Transcriptional Regulators

#### 4.2.1. GATA Binding Protein 1 (GATA1)

GATA1 is a zinc finger transcription factor with a central role in erythropoiesis and acts by modulating complexes with TAL1, LMO2, LDB1, RUNX1, ETO, and ETS family proteins [22]. GATA1 undergoes several posttranslational modifications, including phosphorylation, resulting mainly from EPO activation [30]. Recent AEL patient transcriptomic data show alterations of transcription or downstream signaling factors that mediate GATA1 activity in more than 25% of the cases [31]. Ectopic expression of these physical or functional interactors of the GATA1 transcriptional complexes (ERG, ETO2, SKI, and SPI1) in murine erythroid progenitors resulted in decreased chromatin accessibility at GATA1-binding sites and promoted proliferation with the immature phenotype [31]. Mouse models have confirmed that reduced GATA1 expression, as well as ectopic expression of GATA1 interactors [32,33,34], can promote erythroleukemic phenotypes [35]. Conversely, other AEL mouse models exhibited upregulated erythroid transcription factors (GATA1, FOG-1, and KLF1) and erythroid chromatin access, resulting in ectopic erythroid potential [36]. Mutational analyses of AEL have enlightened, even though rare, the occurrence of fusion genes or mutations affecting GATA1 itself or proteins of the GATA1 complexes (e.g., NFIA-ETO2 and MYB-GATA1) [37,38,39,40,41,42]. Mouse AEL tumors established by CRISPR/Cas9 of HSPCs with *Trp53* and *Bcor* mutations had a gene expression profile recapitulating human AEL tumors with an overexpression of erythroid transcription factors such as *Gata1*, *Gata2*, and *Klf1* [25]. Overall, these findings support the main role of altered GATA1 activity in AEL molecular biology through the diverse effects of erythroid favoritism and the inhibition of normal erythroid differentiation [22].

#### 4.2.2. ETS Transcription Factors (ERG, SPI1, and FLI1)

The E-twenty-six/E26 (ETS) family genes were originally discovered within the erythroleukemia-causing avian retrovirus E26 through the viral *ets* (*v-ets*) oncogene, which was found to have been transduced from homologous genes in the chicken genome to encode part of a hybrid viral protein [43]. ETS transcription factors contain a highly conserved ETS DNA-binding domain that interacts together with other transcription factors to enhance the elements [44]. ERG cooperates with GATA1 to regulate the main hematopoietic transcription factors, such as SCL/TAL1 [45], and is known to promote hemopoietic stem cell (HSC) maintenance and control erythromegakaryocytic differentiation [46]. Higher ERG levels have been related to an unfavorable prognosis in AML [47], and ERG cooperation with GATA1 was found to immortalize hematopoietic progenitor cells [48], further indicating its role in AEL progression. Another ETS transcription factor, SPI1, overexpressed in transgenic mouse models, induced hepatosplenomegaly with erythroblast infiltration and tumor cells in peripheral blood. Nevertheless, these malignant proerythroblasts were partially blocked in differentiation and strictly dependent on erythropoietin for their proliferation both in vivo and in vitro [32]. Interestingly, SPI1 is also known to have a role in HSC maintenance [49]. The *FLI1* gene, encoding for another ETS transcription factor, was also first studied in erythroleukemia virus models, and its overexpression reduces GATA1 expression and impairs erythroid differentiation [50]. FLI1 expression is upregulated by SPI1, implying a synergistic action in AEL [50]. Moreover, engineering mice with an inducible expression of the fusion EWS/FLI-1 resulted in the rapid development of erythroleukemia expressing GATA1 [51]. Mouse models with ectopic expression of ERG, SPI1, and FLI1 confirm that these transcription factors can induce AEL phenotypes [32,33,34], converging on the GATA1 lead to the development of the disease.

#### 4.2.3. Caudal-Type Homeobox 4 (CDX4)

The caudal-type homeobox family consists of CDX1, CDX2, and CDX4, which are developmental regulators of HOX gene expression [52]. Essentially, CDX4 is expressed normally in early hematopoietic progenitors; however, it is expressed aberrantly in around 25% of AML patient samples [53]. In a retroviral transduction/bone marrow transplant model, the onset of AML in mice, induced by MLL-AF9, was substantially delayed when *CDX4* was absent [54]. In vitro, the retroviral overexpression of *CDX4* induced aberrant self-renewal in mice HSC cells, and similarly, in vivo, transplantation induced an AML-like disease in around 50% of mice [53]. This emphasizes that CDX4 is essential for normal hematopoiesis, and its aberrant expression mediates leukemogenesis. However, its direct applicability to human disease remains in question, as a transcriptome analysis of human AEL samples has not identified *CDX4* alterations [31].

### 4.3. Expression of Master Oncogenes

Proto-oncogenes, particularly c-MYC, have a substantial role in erythroleukemia differentiation [55]. This has been studied particularly through mouse models. Murine erythroleukemia (MEL) cells are erythroid progenitors, with halted erythroid differentiation due to transformation with the Friend virus complex. Consequently, they have been employed as a model to explore the molecular biology of cellular differentiation, since DMSO induces terminal erythroid differentiation in these cells [55]. Erythroid differentiation in MEL cells is associated with a decreased expression of the MYC proto-oncogene, while MYC overexpression inhibits differentiation [56]. In vivo, Leder et al. generated transgenic mice harboring the human c-MYC proto-oncogene under the control of key mouse GATA-1 regulatory sequences. Tumor cells displayed proerythroblast morphology and expressed erythroid lineage markers, including EPOR and β-globin [57]. Thus, aberrant MYC activation at a vulnerable phase of erythroid differentiation likely triggers erythroleukemia. Additionally, *H-Ras* and *K-Ras* oncogenes are upregulated in MEL cells [58]. Leder et al. developed another transgenic mouse model (Tg.AC) in which the embryonic zeta-globin promoter was fused to the v-Ha-RAS oncogene, driving its expression [59,60]. These transgenic mice developed multiple mesenchymal and epithelial neoplasms, with a few (estimated to be <5%) developing hepatosplenomegaly with erythroblast infiltration [61]. Peripheral blood results showed a marked increase in metarubricytes and other less differentiated erythroid progenitor cells, including leukemic cells stained positive for GATA-1 [61]. Similarly, in vitro, EPO-induced differentiation was inhibited when a constitutively active RAS mutant (RAS12V) was expressed in SKT6 cells, a Friend murine erythroleukemic cell line. This suggests that aberrant RAS activation can drive erythroid transformation.

### 4.4. Impaired TP53 Activity in AEL Biology

AEL is characterized by a high prevalence of biallelic *TP53* mutations in both de novo and secondary cases [14,20]. The ICC especially emphasizes this feature, as its 2022 definition of AEL stipulates a *TP53* mutation as a required characteristic to establish a diagnosis. Most recent studies have similarly noted the association of *TP53* mutation with AEL.

A retrospective study of 92 patients by Grossman et al. found that *TP53* was mutated in only 43.5% of patient cases [20]. Another study of 58 AEL patients only reported *TP53* mutations to be present in 12 patients (36.3%). This study also demonstrated these individuals to have an average of 4.41 mutations per sample, as well as a higher cytogenetic risk and poorer outcome. These studies classified patients according to the less specific 2008 WHO criteria, which could explain the lower proportion of *TP53*-mutated AEL in their cohort [31].

More recently, studies have begun using the 2016 and 2022 WHO criteria for AEL classification. In a retrospective analysis by the Mayo Clinic, biallelic *TP53* alterations were present in all 41 (100%) AEL cases [10]. Similarly, a study published by the MD Anderson Cancer Center group also reported 100% incidence of *TP53* mutations, most commonly a mutation in one allele and deletion in the other allele in 21 AEL cases [14]. These newer, specific, and stringent definitions for AEL appear to be associated with a higher rate of *TP53* mutations. As such, it is important to note which criteria for AEL diagnosis is being used when evaluating previous studies.

*TP53* mutations represent by far the most common genomic alteration in AEL; however, the specifics of functional involvement in erythroid differentiation are not yet thoroughly clarified [22]. During normal erythropoiesis, p53 and GATA1 were found to interact through GATA1 DNA-binding domains to promote erythroid cell development and survival [62]. Moreover, recently, p53 activation during ribosome biogenesis was found to regulate normal erythropoiesis [63]. *TP53* mutations in AEL and other subtype AML patients promote the proliferation and survival of hematopoietic stem cells and progenitor cells (HSPCs), accumulating additional DNA damage, and are associated with poor prognosis [64,65]. Nevertheless, at least 80% of *TP53*-mutated AML patients present more than one genetic alteration, suggesting that additional abnormalities are required for development of the disease [66]. Indeed, *TP53* mutations cooperate with multiple alternate pathways to produce the AEL phenotype. The RAS signaling pathway is shown to cooperate with impaired p53 activity, as *KRAS* and *NRAS* mutations are shown to produce AML and AEL phenotypes, respectively, when combined with the loss of p53 activity in mice [67,68]. Recent transcriptomic analyses showed that, in *TP53*-mutated AEL samples, transcription factor ERG is upregulated, and transplanting purified ERG-transduced *TP53*-mutated HSPC erythroblasts resulted in fatal erythroleukemia within 60 days [31]. Thus, *TP53* mutations can cooperate with high ERG expression to enhance the proliferation of erythroid progenitors and development of AEL. Similarly, various other genetic lesions have been found to cooperate with *TP53* loss of function mutations, such as *JAK2V617F*, *NTRK1H498R*, the t(1;16)(p31;q24) chromosomal translocation (*NFIA-ETO2* fusion gene), and loss of function mutations of *BCOR* with or without *DNMT3A* mutations [25,37,69,70,71].

Additionally, the analysis of edited sites in leukemia mice models established by CRISPR/Cas 9 genome editing showed that concomitant *TP53* and *Bcor* mutations are central drivers of erythroleukemia [25]. These *TP53*-mutated tumors also acquired secondary mutations in signaling pathway genes, including *Ptpn11*, *Kit*, *Kras*, *Nras*, and *Csf1r1*, in addition to cell cycle regulators and DNA repair genes [25]. Meanwhile, bone marrow samples from mice that did not develop tumors were enriched with gRNAs targeting *Tet2*, *Dnmt3a*, *Stag2*, and *Asxl1* but not *Nfx1*, *Rb1*, *TP53*, or *Bcor* (apart from one mouse). The latter further emphasizes the essential role of *TP53* and *Bcor* co-mutation, as mutations in other listed genes alone were insufficient to drive leukemogenesis [25].

### 4.5. Impaired C/EBPα Function

#### GATA2 and C\EBPα

GATA2 and C\EBPα are two essential transcription factors involved in hematopoiesis. On their role in AEL, there was a reported significant association between *GATA2* mutations and biallelic *C\EBPα* mutations in a group of 55 AEL patients [72]. In fact, biallelic *C\EBP* and *GATA-2* zinc finger 1 (ZF1) mutations synergize in leukemia progression [73]. This was also evident by a mouse model that showed biallelic *C\ebpα* led to myeloid leukemia development, and the addition of *Gata2* mutation to the latter promoted leukemia progression, with 40% of triple transgenic mice developing leukemia with both erythroid and myeloid features. Biallelic *C\ebpα* mutations enhanced erythroid genes expression, while *Gata2* mutations induced chromatin accessibility at erythroid transcription factor motifs such as *Gata1*, *Zfpm1*, and *Klf1* and reduced it at myeloid transcription factor motifs [36]. In addition, published data on *GATA2* mutants revealed that there is an increased expression of erythroid-related antigens Ter-119, β-globin, and βh1-globin and increased hemoglobin positivity in GATA2 mutants compared to controls [72]. The above findings confirm that mutant GATA2 activity controls the abnormal chromatin accessibility at crucial loci regulated by erythroid transcription factors, leading to this erythroid phenotype.

### 4.6. Epigenetic Dysregulation in Erythroleukemia

AEL is also associated with other somatic mutations in epigenetic and transcriptional regulators. Fagnan et al. found that a significant percentage of AEL patients carry mutations in genes encoding epigenetic regulators (33.3% of the cohort). These were mainly *TET2* nonsense mutations (*n* = 8) and *DNMT3A* (*n* = 5) mutations [31]. The samples had an average of 5.72 mutations. Several patients carrying *TET2* and *DNMT3A* mutations also had mutations in *SRSF2* or *IDH2.* DNA methylation plays an essential role in erythroid malignancies, and it is regulated by several factors, including TET2 and DNMT3A/B [74,75]. In mice models, inactivating mutations of *Tet2* and *Dnmt3a/b* promotes hematopoietic stem cell (HSC) renewal and inhibits differentiation, leading to leukemic transformation [76,77]. In HSCs, inactivation of these two genes led to an increase in myeloid and decrease in erythroid gene expression, with erythroid progenitors accumulating in mice. Similarly, in human AEL samples, *DNMT3A/B* and/or *TET2* mutations were present in some AEL patients, and the combination of *TET2* and *DNMT3A* was reported to upregulate erythroid transcription factor KLF1 and EPOR in HSCs [31]. Recently, changes in the promoter methylation and gene expression of erythroid transcription factors (GATA1, KLF1, TAL1, JAK2, etc.), as well as factors implicated in protein binding (MEF2c, BRAF, RCOR1, LIFR, and CTNNA1), were observed in AEL mouse models. Moreover, *Dnmt3a*-mutated/*Tet2* wild-type leukemia models, or *Dnmt3a*-mutant/*Tet2* heterozygous models, were of the erythroid phenotype and exhibited hypomethylation and overexpression of genes regulating erythrocyte differentiation and homeostasis, including *Gata1*, *Klf1*, and *Kit*, and hypermethylation and low expression of genes involved in leukocyte activation and differentiation (e.g., *Myb*, *Tnfaip3*, *Ikzf3*, and *Cd74*). These recent data further highlight the role of DNA methylation in AEL molecular biology [25] and support the use of cytidine analogs such as 5-Azacytidine or Decitabine as potential therapeutic options [78].

### 4.7. Other Less Frequent Genetic Alterations

The study by Fagnan et al. also found a subset of AEL cases (*n* = 10, or 30.4% of their cohort) with a significantly lower mutational burden, having less mutations per sample on average than *TP53*-mutated or epigenetic regulator subgroups. These AEL patients had an average of 1.60 mutations per sample, which was significantly lower than their *TP53*-mutated or epigenetic regulator subgroups [31]. As this study used the 2008 WHO definition of AEL, some of their samples may be categorized differently than the newer classification models.

In a 2013 study, Grossman et al. found that *NPM1* was mutated in 15 out of 92 patients (16.3%). Other less frequent mutations noted in this study were mutations in *ASXL1* (8.0%), *RUNX1* (8.7%), *MLL-PTD* (7.8%), *IDH1* (7.5%), *IDH2* (4.7%), *NRAS* (3.3%), *KRAS* (3.3%), *FLT3-ITD* (3.3%), *FLT3-TKD* (3.5%), *SF3B1* (2.5%), and *CEBPA* (1.1%) [20]. This study demonstrated that the mutation load was similar across all of the gene mutations analyzed [20]. Again, it is important to note that this study used the old definitions of AEL and that mutation frequencies may differ according to the new criteria. The recent analysis of 41 AEL patients by the Mayo Clinic with the updated AEL definitions demonstrated that 20% of patients had mutations in *TET2* and 10% of patients had mutations in *ASXL1*, *IDH2*, and *DNMT3A* [10].

In addition, the *NUP98* gene encodes a nucleoporin protein that acts as a transcription activator and has been linked to over 28 hematologic malignancies, including AEL [79]. When analyzing a cohort of pediatric cases, *NUP98* fusions appeared to be highly enriched in AEL patients compared to those with other AML types (31.8 vs. 6.7%) [80]. This percentage was even higher in the AEL category, as three out of five patients (60%) exhibited this fusion. Of the seven total patients with *NUP98* fusions, four had *KDM5A* as the fusion partner, two had *NSD1*, and one had *SET* [80]. Those with this *NUP98* fusion appeared to have higher OS scores; however, this finding was not statistically significant due to the limited sample size (OS = 45 ± 28 no fusion, OS = 65 ± 33 with *NUP98* fusion).

## 5. Treatments and Clinical Outcomes

Due to the rarity of AEL diagnosis and the relatively poor understanding of the disease biology, treatment options are limited at this time (Table 1). The current first-line treatment strategies include intensive chemotherapy (ICT) and hypomethylating agents (HMAs). Allogeneic bone marrow transplant (AlloBMT) is the only potential curative approach, but it requires deep remission prior to its initiation, which proves challenging in this disease.

### 5.1. Intensive Chemotherapy

ICT is often used as a front-line treatment for AEL patients who are eligible for intensive therapy [3]. In a large retrospective multinational study of 217 patients with AEL by Almeida et al., 122 patients were treated with ICT. Of the 119 whose response data was available, daunorubicin (45 or 60 mg/m^2^ × 3 days) with cytarabine (100 mg/m^2^ bid × 7 days) was the most frequently used induction regimen (*n* = 81, 66%). Other regimens used included idarubicin 12 mg/m^2^ (*n* = 25, 20%) or mitoxantrone 12 mg/m^2^ (*n* = 8, 7%) in combination with cytarabine. The ICT group had a median age of 60 at diagnosis, compared to 69 for the HMA group. The objective response rate (ORR) was 72%, according to the ELN criteria. Complete response (CR) occurred in 79 patients (66%), partial response (PR) in 7 (6%), stable disease (SD) in 16 (13%), and primary disease progression (PPD) in 17 (14%). Following treatment with ICT, 23 (18.8%) patients received an allogeneic bone marrow transplant. Patients treated with ICT followed by a transplant experienced a median OS of 5.9 months. In the ICT group overall, the authors reported a median OS of 10.5 months but did not find a significant difference in OS between IHMA-treated and ICT-treated patients with MRC intermediate-risk cytogenetics (29.3 vs. 16.9 months, *p* = 0.277). In patients with MRC high cytogenetic risk, the median OS was significantly higher in the first-line HMA group compared to ICT (13.3 vs. 7.5 months, *p* = 0.039) [3]. However, as the definitions have changed to become more stringent, these data may not be as relevant to AEL as it is currently defined, and many of these patients would likely be classified as MDS with excess blasts or AML.

Treatment with chemotherapy and younger age have also been associated with significantly higher overall survival in AEL [3,15]. A recent 2023 retrospective analysis by Gera et al. demonstrated an inverse relationship between age and OS in patients with AEL, with a median OS of 69, 18, 8, 3, and 1 month for age groups <18, 18–49, 50–64, 65–79, and 80+, respectively [15]. Pediatric AEL shows a distinct genomic profile, which could partially explain the more favorable outcomes in this age group [15]. Additionally, in many hematologic malignancies, the pediatric chemotherapy regimen is more aggressive, as children are better able to tolerate this treatment compared to adults [19]. While the effect of chemotherapy is heightened amongst pediatric patients, the use of chemotherapy appears to be associated with increased survival in both pediatric and adult populations. The OS in this study was 152 vs. 2 months in children who received chemotherapy vs. those who did not, and the OS was 8 vs. 1 month in adults receiving chemotherapy vs. not [15]. It should be noted that the retrospective nature of these analyses mitigates the significance of these conclusions.

### 5.2. Hypomethylating Agents

The HMAs azacitidine and decitabine are commonly used in AML patients who are not eligible for ICT [15]. HMAs work by reversing the DNA methylation that often silences tumor suppressor genes involved in cancer pathogenesis signaling. Although HMAs can induce a promising initial response in a subset of AEL patients, resistance emerges in nearly all the patients with this disease [82]. This can either occur through primary resistance, in which there is no improvement after four to six cycles of HMAs, or secondary resistance, in which patients progress after an initial response to HMA [82]. Almeida et al. also found that the median progression-free survival (PFS) was longer for first-line HMA treatment compared to second-line or later HMA (9.4 vs. 3.4 months, respectively) [3], although the AEL definition has changed since the study was published.

However, a recent retrospective study of 41 patients with AEL described no benefit to using one treatment regimen over another. The regimens reported in this study included HMA alone, HMA with venetoclax, ICT, and best supportive care [10]. As such, more studies need to be conducted to better understand the varying degrees of efficacy amongst HMA treatments for AEL as defined by the newer 2022 criteria.

#### Venetoclax

Venetoclax is a B-cell lymphoma 2 (BCL-2) inhibitor that has been shown to improve the remission rates and OS of patients with AML who are ineligible for ICT [83]. However, erythroid/megakaryocytic AML subtypes are associated with resistance to venetoclax [84]. This is presumably driven by erythroid/megakaryocytic differentiation, in addition to *TP53* mutations. p53 loss has been linked to venetoclax resistance, along with a concomitant compensatory BCL-XL upregulation [84]. In fact, Kuusanmäki et al. recently found that AML cells exhibiting erythroid/megakaryocytic differentiation depend on BCL-XL rather than BCL-2 for their survival [84]. Thus, BCL-2 inhibition can exhibit limited therapeutic efficacy in AEL.

### 5.3. Allogeneic Bone Marrow Transplant

Allogeneic bone marrow transplantation (AlloBMT) is the only potentially curative approach for AEL, but it requires deep remission of the disease, which is rarely achieved in these patients [15]. It has been previously found that the median survival of AEL patients who received AlloBMT was 66 months from transplant [81]. However, it should be noted that the definition of AEL for this analysis was based on the 2008 WHO classification. The median OS of AlloBMT recipients was 89 months, compared to 5 months for those who did not undergo AlloBMT [81]. In one study, twelve patients underwent AlloBMT, representing 28% of the AEL cohort, and 71% of those were in complete remission (CR1 or CR2) [81]. Thus, using AlloBMT as a consolidation therapy may improve outcomes in AEL. A successful initial treatment to achieve remission is currently a requirement to undergo AlloBMT. This stipulation remains a challenge, as only a small proportion of patients reach AlloBMT.

### 5.4. Chimeric Antigen Receptor T-Cell Therapy

Genetically engineered T cells, such as chimeric antigen receptor (CAR) T cells, have started to be used therapeutically in cancers such as leukemias and lymphomas by targeting distinct antigens present on malignant cells [85,86]. Recently, some studies have examined potential targets of CAR-T therapy in AML. In a 2023 study, Gottschlich et al. investigated RNA sequencing data of single cells from individuals with AML and healthy tissue to determine epitopes expressed selectively on malignant cells. Through computational analysis and subsequent validation, the authors found that colony-stimulating factor 1 receptor and cluster of differentiation 86 could be used as CAR-T therapy targets in AML [85]. In 2019, Gomes-Silva and colleagues found that CD7, which is expressed by the malignant blasts and progenitor cells of a subset of AML patients, acted as a target for CAR-T therapy [87]. Overall, the variety of molecular epitopes that have been identified in CAR-T therapy for AML could represent beneficial future applications for single-cell cancer treatment.

While there have been promising advances in CAR-T cell therapy for the treatment of some leukemias, including AML, studies investigating single-cell analysis in AEL remain limited. As clinical outcomes for AEL are poor and there are few treatment options, CAR-T therapy should be further investigated as a treatment modality for AEL.

### 5.5. Future Directions

Given the overall poor outcomes of AEL patients, it is evident that clinical trials represent the best option for these individuals. The current trials aimed at treating AEL patients specifically are limited due to the rarity of this condition, especially with the newest definitions and classifications.

Additionally, it was recently reported that the survival rates for AEL have not improved over the last twenty years, despite the introduction of novel targeted therapies for the treatment of AML and the advancements made in the process of AlloBMT resulting in improved non-relapse mortality [15]. It is paramount to continue seeking novel therapeutic advancements to improve these clinical outcomes. Among the challenges is that the main drivers of this disease appear to be transcriptional factors that are not easily targetable and that JAK2 inhibition alone does not seem to be able to control this disease or change its natural history.

Recently, Iacobucci et al. demonstrated a sensitivity to CDK7 and CDK9 inhibition for *TP53*-, *BCOR*-, and *DNMT3A*-mutated AEL models and further confirmed CDK9 efficacy in AEL samples with combinatorial mutations in *TP53*, *BCOR*, *NFIX*, and *RB1* [25]. Furthermore, the authors found that *TP53*-mutated AEL tumors were shown to have high sensitivity to PARP inhibition (talazoparib) in the absence of *DNMT3A* mutation [25]. As drugs such as decitabine increase the presence of PARP1 at DNA damage sites, this amplifies the cytotoxic effects and, thus, could represent an effective targeted therapy for *TP53*-mutated AEL [25,88]. Lastly, the efficacy of gemcitabine and bromodomain inhibitors was demonstrated in *NUP98::KDM5A* leukemia [25].

Furthermore, the molecule BCL-XL has shown promise as a therapeutic target in AEL. BCL-XL is an antiapoptotic protein that has been shown through CRISPR screens to exhibit high expression in erythroid/megakaryocytic AML, thus conferring their survival [84]. A 2023 study by Kuusanmäki and colleagues found BCL-XL-selective inhibitors, rather than BCL-2 inhibitors such as venetoclax, to be highly effective against erythroid/megakaryoblastic leukemia cell lines. When these inhibitors were combined with the *JAK* inhibitor ruxolitinib, the cell lines showed a synergistic response to the therapy [84]. Moreover, it was demonstrated ex vivo that BCL-XL inhibition successfully eliminated blasts from patients with AML who had erythroid or megakaryocytic differentiation, as well as decreased tumor burden, in a mouse model of an erythroleukemia xenograft [84]. A recent review discussed that navitoclax, which specifically inhibits BCL-XL and BCL-2, displayed therapeutic benefit and antineoplastic effects [89,90]. The combined use of BCL-XL inhibitors with chemotherapy has also been shown to be an effective treatment against acute leukemia [89,91]. As such, compounds that target the BCL-XL molecule may be promising in the future of AEL treatment.

*JAK2* inhibition also demonstrates clinical utility in the treatment of AEL. A recent study by Li and colleagues found that biallelic *TP53* inactivation leads to leukemic transformation in PEL and that *JAK2/TRP53*-mutated PEL exhibits DNA damage and persistent copy number variations [92]. The authors further demonstrated that PEL demonstrated high sensitivity to the inhibition of molecules involved in DNA repair, such as WEE1 and poly(ADP-ribose) polymerase (PARP), which could offer a prospective therapeutic approach to this disease [92]. Another recent study found a high frequency of gains and amplifications encompassing *EPOR/JAK2* in *TP53*-mutated cases of AEL, which were frequently associated with poor prognosis [29]. However, these samples exhibited a high sensitivity to ruxolitinib, a *JAK2* inhibitor, in both in vitro and xenograft models. This further supports the potential role of *JAK2* inhibition in the targeted treatment of AEL cases with altered JAK2 signaling.

Examining future prospects, there are also ongoing and recently completed clinical trials examining the sensitivity of cohesin-mutated AML and MDS with excess blasts to talazoparib (#NCT03974217), as well as evaluating the use of gemtuzumab ozogamicin in CD33^+^ relapsed or refractory AML (#NCT04207190) [25]. A recent phase I clinical trial studied the use of decitabine with talazoparib in 25 patients with refractory/relapsed AML and found complete remission with incomplete count recovery in 8% (*n* = 2), as well as hematologic improvement in 12% (*n* = 3) [93]. Clinical trials investigating drug responses in patients with AEL remain limited, although recent AEL models have shown promising responses [25,84,92]. Thus, future directions include targeting AEL based on a better understanding of the molecular biology of the disease, as well as the confirmation of preclinical findings using patient samples.

## 6. Conclusions

Overall, AEL is a very aggressive and rare disorder that is associated with poor clinical outcomes. The WHO and ICC differ slightly on their nomenclature and criteria for defining this condition; however, both acknowledge the presence of the hypercellularity of the bone marrow and predominance of erythroblast precursors, as well as a supporting *TP53* mutation with evidence of biallelic inactivation [6,11]. In addition to bone marrow biopsy, next-generation sequencing, flow cytometry, and immunohistochemistry are all supporting measures used to aid the diagnosis of AEL [1].

Because of the high-risk features of this disease, clinical outcomes remain poor. HMAs, intensive chemotherapy, and AlloBMT have all been used, with overall limited efficacy. Younger age and the use of chemotherapy aid in survival time; however, AlloBMT is the only potentially curative approach for this disease [3,15].

In conclusion, the improvement of the outcomes of this rare and aggressive disease requires a better understanding of its molecular biology, introduction of novel therapeutic strategies, and treatment of these patients in multi-institutional clinical trials that will allow the evaluation of therapies in enlarged cohorts. These next steps remain challenging given the stricter definitions of AEL.

## Figures and Tables

**Figure 1 ijms-25-06256-f001:**
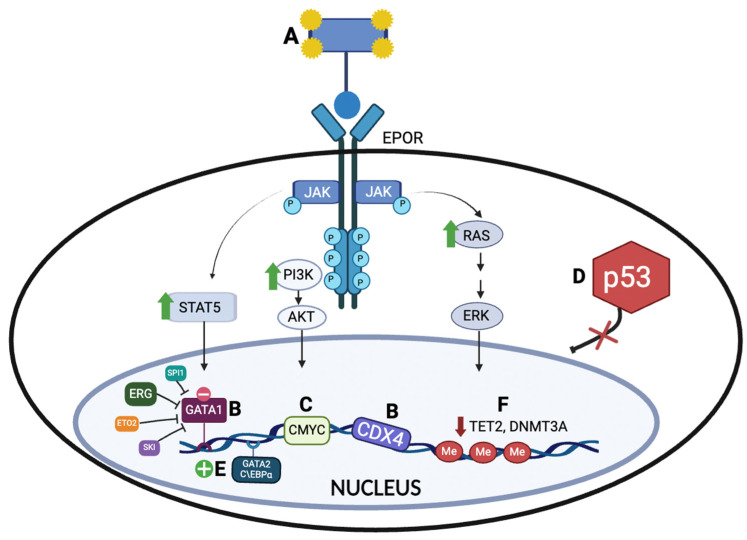
Mechanisms implicated in the pathobiology of AEL. (**A**) EPOR activation and subsequent constitutional activation of the downstream signaling effectors, such as signal transcription activators (STATs), PI3K/AKT, and mitogen-activated protein (MAP) or extracellular signal-regulated (ERK) kinases, promote the development of proerythroblasts. (**B**) Reduced *GATA1* expression, as well as the ectopic expression of GATA1 interactors/transcriptional complexes (ERG, ETO2, SKI, and SPI1), results in the inhibition of normal erythroid differentiation. An aberrant overexpression of *CDX4*, a developmental regulator, mediates leukemogenesis. (**C**) Overexpression of a proto-oncogene, *c-MYC*, also halts erythroid differentiation. (**D**) *TP53* mutations promote the proliferation and survival of hematopoietic stem cells and progenitor cells, accumulating additional DNA damage. (**E**) Activating mutations in *GATA2* and C/EBPα, enhancing both erythroid genes expression and chromatin accessibility. (**F**) Epigenetic dysregulation (inactivating mutations of *TET2* and *DNMT3A/B*) promote hematopoietic stem cell renewal and inhibit differentiation. Created with BioRender.com.

**Table 1 ijms-25-06256-t001:** Previous clinical studies on Acute Erythroid Leukemia and their respective outcomes.

Authors	AEL Definition Criteria	Participants	Treatment	Outcome	Reference Number
Almeida et al., 2017	WHO 2008 Criteria for AEL	217 total patients with AEL pooled from 28 international registries and 8 countries (1998–2014)	HMA or ICT	Median OS: 11.1 mo Median PFS: 7.1 mo 1-Year Survival: 49% No significant difference between OS between groups High risk cytogenic group had higher survival rate with HMA (7.5% vs. 13.3%, *p* = 0.039)	[3]
		88 patients with AEL (mean age 69)	HMA 66 primary, 11 secondary, 11 unknown	Median OS= 13.7 mo PFS was longer for first line HMA vs. second-line/later (9.4 vs. 3.4 months) 1 year survival: 65.8%	[3]
		122 patients with AEL (mean age 60)	ICT 81 primary, 17 secondary, 24 unknown daunorubicin with cytarabine (N = 81) idarubicin with cytarabine (*n* = 25) mitoxantrone with cytarabine (*n* = 8)	Median OS = 10.5 mo Median PFS: 8.0 mo 1-Year survival: 46.7%	[3]
Gera et al., 2023	WHO 2001 Criteria for AEL	968 patients with PEL from the 2000–2019 SEER database (Median Age 68 years old, 62% male)	65% of patients were treated with ICT	Patients who received ICT had an OS was significantly higher, adults *p* < 0.0001, children *p* = 0.004 Median OS revealed no change from 2000–2019 OS was significantly correlated with younger age (*p* < 0.0001)	[15]
		918 Adults > 18 years of age	559 patients treated with ICT	Adults Median OS: 5 mo 5-Year survival: 9.47 mo	[15]
		50 Children < 18 years of age	46 patients treated with ICT	Children Median OS: 69 mo 5-Year survival: 55.01%	[15]
Reichard et al., 2022	WHO 2016 Criteria for AEL	41 PEL patients (14 de novo, 12 secondary to MDS, 14 therapy-related) (Mean age 66 years, 71% male)	29 patients had treatment data recordedHMA (*n* = 5), HMA with Venetoclax (*n* = 12), ICT (*n* = 4), and best supportive care (*n* = 8)	All cases expressed biallelic TP53 mutations Overall mean OS: 3.3 mo (Median 2 mo) ○ Of the 29 patient subgroup, mean OS = 1.8 mo	[10]
Alkhateeb et al., 2016	WHO 2008 Criteria for AEL	43 patients at Mayo Clinic 12 underwent HCT (Median Age 65, 79% male)	Stem Cell Transplant 1 Autologous 11 Allogenic 3 MRD 8MUD Conditioning regimen 6 Flu/Mel 1 Flu/Blu 5 Cy TBI	Median OS: 66 mo 50% acute GHVD, 33% Chronic GHVD Median OS HCT 89 months, no HCT 5 months (*p* = 0.003)	[81]

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
