# Peer review of "Acute Erythroid Leukemia: From Molecular Biology to Clinical Outcomes"

_ijms, 2024, doi:10.3390/ijms25116256_

Round 1
Reviewer 1 Report
Comments and Suggestions for Authors
This review describes the current changes in the classification criteria of acute erythroid leukemia, the genetic and biological features, the available treatment options and associated clinical outcome. The diagnosis of AEL is still controversial due to continuous changes in the classification criteria but important given its dismal outcome. Although the authors described the major changes from both the WHO 5th edition and the ICC classifications, it is not clear which definition or criteria the authors used with the term “AEL” throughout the text. This should be clarified at the beginning of the introduction. It seems that in most cases the authors use “AEL” for “PEL”, pure erythroid leukemia.
Major comments
In the paragraph 2.1 which described the classification changes according to the WHO, the authors should mention the historical distinction between the initial FAB M6a and M6b and the following partitioning of AEL from the WHO into an erythroid/myeloid (M6a) subtype (requiring ≥50% erythroid precursors and ≥20% myeloblasts within the total nucleated fraction) and a pure erythroid (M6b) subtype (requiring >80% erythroid precursors and ≥30% proerythroblasts). The erythroid/myeloid (M6a) disappeared from the classification in the WHO revision of 2016, mostly replaced by AML-MRC and MDS.
Page 2, lines 85-86: “ The 2022 International Census Classification (ICC) system took both blast percentage and genomic factors into account when discussing the diagnosis of AEL and termed this condition as AML with TP53 mutation rather than as an erythroid leukemia”. This sentence should be re-phrased since it is not entirely correct. It is true that the association with biallelic TP53 defects and complex karyotype has been emphasized, however cases defined “AML with TP53 mutations” may also include non AEL cases. This should be clarified. Moreover, the association with both TP53 mutations and complex karyotype most commonly refers to pure erythroid leukemia cases.
The authors describe TP53 mutations both in the “Genetic features” paragraph and in the “Molecular Biology” session. Can the authors condensate these parts into one paragraph? Since there is not a specific genetic alteration of AEL, the “Genomic features” paragraph should be discussed within the session “Molecular Biology of ALL”.
Very recent results from CRISPR/Cas9 dependency studies/high throughput drug screening in both mouse models and human cell lines of AEL have not been discussed. However, these are important since provide novel targets for therapeutic inhibition.
Minor
Page 2, line 59, “Giovanni” is misspelt. Please correct it.
Comments on the Quality of English Language
Moderate revision of English language required
Author Response
Reviewer 1
This review describes the current changes in the classification criteria of acute erythroid leukemia, the genetic and biological features, the available treatment options and associated clinical outcome. The diagnosis of AEL is still controversial due to continuous changes in the classification criteria but important given its dismal outcome. Although the authors described the major changes from both the WHO 5th edition and the ICC classifications, it is not clear which definition or criteria the authors used with the term “AEL” throughout the text. This should be clarified at the beginning of the introduction. It seems that in most cases the authors use “AEL” for “PEL”, pure erythroid leukemia.
We appreciate the valuable feedback provided by the reviewer regarding the clarity of our terminology in the manuscript and we regret any confusion caused. To address this, we have added a sentence in the introduction clarifying that we will be using the term “AEL” based on the most recent 2022 WHO terminology in our paper (lines 54-56) and have made the terminology consistent throughout the text. The sentence clarifying our review’s AEL terminology was included when we first begin discussing how various classification naming conventions exist.
While our paper uses the WHO 2022 criteria for the definition when discussing AEL, many of the papers we cite use various definition criteria for defining this disease as the naming conventions and criteria have shifted throughout the past few years. In Table 1, we summarized the studies we cited in the body of our paper and made note of which definition criteria they used when selecting patients for their respective papers.
Major comments
In the paragraph 2.1 which described the classification changes according to the WHO, the authors should mention the historical distinction between the initial FAB M6a and M6b and the following partitioning of AEL from the WHO into an erythroid/myeloid (M6a) subtype (requiring ≥50% erythroid precursors and ≥20% myeloblasts within the total nucleated fraction) and a pure erythroid (M6b) subtype (requiring >80% erythroid precursors and ≥30% proerythroblasts). The erythroid/myeloid (M6a) disappeared from the classification in the WHO revision of 2016, mostly replaced by AML-MRC and MDS.
We thank the reviewer for this helpful feedback on the historical changes to the classification of AEL over time. We agree that this historical piece is important and helps better connect the previous naming conventions to the current ones. As such, we have made extensive changes to the definition paragraphs to incorporate the distinction between M6a and M6b made in 2001 by the WHO as well as the disappearance of the M6a criteria from the 2016 WHO classification (lines 72-82).
Page 2, lines 85-86: “The 2022 International Census Classification (ICC) system took both blast percentage and genomic factors into account when discussing the diagnosis of AEL and termed this condition as AML with TP53 mutation rather than as an erythroid leukemia”. This sentence should be re-phrased since it is not entirely correct. It is true that the association with biallelic TP53 defects and complex karyotype has been emphasized, however cases defined “AML with TP53 mutations” may also include non AEL cases. This should be clarified. Moreover, the association with both TP53 mutations and complex karyotype most commonly refers to pure erythroid leukemia cases.
We greatly appreciate this comment on the 2022 ICC system definition. We clarified the ICC definition and noted its broad nature in including some non-AEL cases as well (lines 108-109). Additionally, we included the PEL subtype of AML with mutated TP53 and its criteria into the final sentence of this paragraph as well [113-115].
The authors describe TP53 mutations both in the “Genetic features” paragraph and in the “Molecular Biology” session. Can the authors condensate these parts into one paragraph? Since there is not a specific genetic alteration of AEL, the “Genomic features” paragraph should be discussed within the session “Molecular Biology of ALL”.
We thank the reviewer for the valuable feedback regarding the structure of our manuscript. The two parts are now combined under the section “Molecular Biology and Genomic Features of AEL” with the different genetic alterations mentioned within this part.
Very recent results from CRISPR/Cas9 dependency studies/high throughput drug screening in both mouse models and human cell lines of AEL have not been discussed. However, these are important since provide novel targets for therapeutic inhibition.
Thank you very much for bringing this important point to our attention. We greatly appreciate this valuable feedback and regret not incorporating the recent results from CRISPR/Cas9 dependency studies and high throughput drug screening earlier. We agree that these findings are very important. We have incorporated into our “Molecular Biology and Genomic Features of AEL” section the findings of the 2021 study by Iacobucci and colleagues, as well as their findings regarding high throughput drug screening, regarding sensitivity to CDK7 and CDK9 in AEL models. Furthermore, we have included the authors’ findings regarding sensitivity to PARP inhibition and decitabine, as well as recently completed and ongoing clinical trials investigating targeted drug sensitivities in AML. This information can be found in lines 274-276, 314-322, 429-438, 474-486, and 653-672.
Minor
Page 2, line 59, “Giovanni” is misspelt. Please correct it.
We appreciate that you caught this error. We have corrected the incorrect spelling which is now on line 67.

Reviewer 2 Report
Comments and Suggestions for Authors
This is an excellent review article from Fernandel et al. about AEL. The article is very informative, well-organized, and well-described. The article proves to be very useful in informing both clinicians and scientists about the recent developments related to this subkind of AML, with an important focus on the most recent classifications.
Overall, my outlook on the article is very positive, and I was not able to detect major flaws or imperfections. With effort, I was able to spot some very minor issues that I will point here to the authors:
Line 380 and going on: the authors state that inactivation of TET2 and DNMT in HSC leads to an increase in myeloid and a decrease in erythroid gene expression. This has no immediate citation to review this statement, but I reviewed reference #79 and it seems that the concept comes from there. My knowledge, and upon reviewing 79 leads me to believe that the authors meant the opposite, an increase in erythroid and a decrease in myeloid gene expression. If that is not the case, the authors should elaborate more about why this counterintuitive gene expression is happening in the context of AEL.
Line 410: MRC was not defined, to the best of my knowledge.
Line 423: repetition of "better".
A comment on the overall paper: although not much is known, a consideration of cellular therapies such as CAR-T could be helpful, as well as considering the latest single-cell (or lack of) studies.
Author Response
Reviewer 2
This is an excellent review article from Fernandes et al. about AEL. The article is very informative, well-organized, and well-described. The article proves to be very useful in informing both clinicians and scientists about the recent developments related to this subkind of AML, with an important focus on the most recent classifications.
Overall, my outlook on the article is very positive, and I was not able to detect major flaws or imperfections. With effort, I was able to spot some very minor issues that I will point here to the authors:
We thank the reviewer for the kind words.
Line 380 and going on: the authors state that inactivation of TET2 and DNMT in HSC leads to an increase in myeloid and a decrease in erythroid gene expression. This has no immediate citation to review this statement, but I reviewed reference #79 and it seems that the concept comes from there. My knowledge, and upon reviewing 79 leads me to believe that the authors meant the opposite, an increase in erythroid and a decrease in myeloid gene expression. If that is not the case, the authors should elaborate more about why this counterintuitive gene expression is happening in the context of AEL.
We are sorry for not referencing this statement. It is from” Molecular Landscapes and Models of acute erythroleukemia” (https://doi.org/10.1097%2FHS9.0000000000000558) mentioned under the section of “Aberrant chromatin organization and erythroleukemia”. This sentence intends to mean that the inactivation of the two genes led to aberrant accumulation of erythroid progenitors in mice with an increase in myeloid and decrease in erythroid gene expression in hematopoietic stem cells, so these genes were important regulators for hematopoietic differentiation. Their inactivation halts differentiation and leads to progenitors’ accumulation. We tried to remove any confusion and made the sentence clearer now. Thank you for the comment.
Line 410: MRC was not defined, to the best of my knowledge.
We appreciate you alerting us to this error and apologize for this confusion. To address this, we now describe the MRC definition as “myelodysplasia related changes” at the first mention of MRC (line 78).
Line 423: repetition of "better".
Thank you so much for drawing our attention to this repetition. We have deleted the second “better” on line 551.
A comment on the overall paper: although not much is known, a consideration of cellular therapies such as CAR-T could be helpful, as well as considering the latest single-cell (or lack of) studies.
Thank you very much for this feedback, we greatly appreciate this helpful suggestion. We agree that including a consideration of cellular therapies and the most recent single-cell studies would be a valuable addition to this review. To address this, we have added information regarding CAR-T therapy as well as recent CAR-T targets examined in AML. Furthermore, we commented on the limited data surrounding single-cell studies in AEL. This can be found in lines 680-696 (section 5.4).

Round 2
Reviewer 1 Report
Comments and Suggestions for Authors
Although the authors made efforts to address most comments, there are still concerns which would require clarification:
11) Page 3, “2.2 International Census Classification”: this paragraph has wrong statements. The ICC classification does not define AEL as “AML with TP53 mutations” but rather defined a category of “AML with TP53 mutations” which “encompasses separate diagnoses of MDS, MDS/AML, and AML with mutated TP53 (including pure erythroid leukemia), according to the blast percentage. These diseases are grouped together because of their overall similar aggressive behavior irrespective of the blast percentage, warranting a more unified treatment strategy across the blast spectrum” (from PMID: 35767897). This should be corrected.
22) Page 8: lines 316-321: “in the same study, DNA- and RNA-sequencing coupled with flow cytometry of CRISPR/Cas9 engineered AEL tumor models, have demonstrated that the primary clone consists mostly of immature hematopoietic cells expressing erythroid/lymphoid markers and of progenitor erythroid cells expressing GATA1, whereas more differentiated erythroid cells expressing hemoglobin genes are dominant in the secondary tumor passage”. This conclusion is wrong. The paper from reference 25 showed that mouse tumors established by CRISPR/Cas9 of HSPCs and with TP53 and BCOR mutations had a gene expression profile recapitulating human AEL tumors with overexpression of erythroid transcription factors. Based on combinatorial mutation patterns different leukemic phenotypes were established. Please correct.
33) Regarding the therapeutic implications, this Reviewer in the previous Revision referred to the interesting and clinically relevant results about the role of BCL-XL and its inhibition (eg PMID: 36508699 and others) or on the role of DNA repair/JAK2 inhibition (eg. PMID: 35421216). These should be added and discussed.
44) Please check “gene nomenclature” throughout the text since it is not always correct.
Comments on the Quality of English Language
Moderate revision
Author Response
1) Page 3, “2.2 International Census Classification”: this paragraph has wrong statements. The ICC classification does not define AEL as “AML with TP53 mutations” but rather defined a category of “AML with TP53 mutations” which “encompasses separate diagnoses of MDS, MDS/AML, and AML with mutated TP53 (including pure erythroid leukemia), according to the blast percentage. These diseases are grouped together because of their overall similar aggressive behavior irrespective of the blast percentage, warranting a more unified treatment strategy across the blast spectrum” (from PMID: 35767897). This should be corrected.
Thank you for bringing this to our attention. As a response we have revised the wording in section 2.2 to reflect this distinction. Lines 107-110
2) Page 8: lines 316-321: “in the same study, DNA- and RNA-sequencing coupled with flow cytometry of CRISPR/Cas9 engineered AEL tumor models, have demonstrated that the primary clone consists mostly of immature hematopoietic cells expressing erythroid/lymphoid markers and of progenitor erythroid cells expressing GATA1, whereas more differentiated erythroid cells expressing hemoglobin genes are dominant in the secondary tumor passage”. This conclusion is wrong. The paper from reference 25 showed that mouse tumors established by CRISPR/Cas9 of HSPCs and with TP53 and BCOR mutations had a gene expression profile recapitulating human AEL tumors with overexpression of erythroid transcription factors. Based on combinatorial mutation patterns different leukemic phenotypes were established. Please correct.
Thank you for your valuable comment. We have addressed accordingly. Lines 317-327
3) Regarding the therapeutic implications, this Reviewer in the previous Revision referred to the interesting and clinically relevant results about the role of BCL-XL and its inhibition (eg PMID: 36508699 and others) or on the role of DNA repair/JAK2 inhibition (eg. PMID: 35421216). These should be added and discussed.
Thank you very much for this helpful feedback. Previously, we had discussed BCL-XL and its inhibition under Section 5.2.1 Venetoclax. We have moved this part to Section 5.5 Future Directions (lines 669-683) and expanded upon the discussion by adding details about BCL-XL inhibition and incorporating other studies. We have also added information regarding JAK2 inhibition under Section 5.5 Future Directions (lines 694-695), incorporating findings from the article with PMID 35421216, as well as expanding upon the discussion of the article with PMID 35839275.
4) Please check “gene nomenclature” throughout the text since it is not always correct.
Thank you so much for this observation. We have edited the paper to reflect these correted gene nomenclatures.

Round 3
Reviewer 1 Report
Comments and Suggestions for Authors
The manuscript has been improved. However, there are still minor issues.
Lines 241-243: “Mouse AEL tumors established by CRISPR/Cas9 of HSPCs with TP53 and BCOR mutations had a gene expression profile recapitulating human AEL tumors with overexpression of erythroid transcription factors such as GATA1, GATA2 and KLF1”: since these are mouse genes should be: “Mouse AEL tumors established by CRISPR/Cas9 of HSPCs with Trp53 and Bcor mutations had a gene expression profile recapitulating human AEL tumors with overexpression of erythroid transcription factors such as Gata1, Gata2 and Klf1”. Please also check and change accordingly lines 351-360 and throughout the manuscript.
Please check the gene nomenclature again, as some gene names should be in italics. For example, check paragraph 4.6 but also others.
Line 554: NUP98-KDM5A should be NUP98::KDM5A
Comments on the Quality of English Language
Minor revision.
Author Response
The manuscript has been improved. However, there are still minor issues.
Lines 241-243: “Mouse AEL tumors established by CRISPR/Cas9 of HSPCs with TP53 and BCOR mutations had a gene expression profile recapitulating human AEL tumors with overexpression of erythroid transcription factors such as GATA1, GATA2 and KLF1”: since these are mouse genes should be: “Mouse AEL tumors established by CRISPR/Cas9 of HSPCs with Trp53 and Bcor mutations had a gene expression profile recapitulating human AEL tumors with overexpression of erythroid transcription factors such as Gata1, Gata2 and Klf1”. Please also check and change accordingly lines 351-360 and throughout the manuscript.
We thank the reviewers for this point. We have review and correct these points.
Please check the gene nomenclature again, as some gene names should be in italics. For example, check paragraph 4.6 but also others.
We addressed this point as well.
Line 554: NUP98-KDM5A should be NUP98::KDM5A
We corrected this point as well.
